# Ultrafast perturbation maps as a quantitative tool for testing of multi-port photonic devices

Kevin Vynck[1], Nicholas J. Dinsdale [2,3], Bigeng Chen [2], Roman Bruck[2], Ali Z. Khokhar[3], Scott A. Reynolds[3], Lee Crudgington[3], David J. Thomson[3], Graham T. Reed[3], Philippe Lalanne[1] & Otto L. Muskens [2]

Advanced photonic probing techniques are of great importance for the development of non-contact wafer-scale testing of photonic chips. Ultrafast photomodulation has been identified as a powerful new tool capable of remotely mapping photonic devices through a scanning perturbation. Here, we develop photomodulation maps into a quantitative technique through a general and rigorous method based on Lorentz reciprocity that allows the prediction of transmittance perturbation maps for arbitrary linear photonic systems with great accuracy and minimal computational cost. Excellent agreement is obtained between predicted and experimental maps of various optical multimode-interference devices, thereby allowing direct comparison of a device under test with a physical model of an ideal design structure. In addition to constituting a promising route for optical testing in photonics manufacturing, ultrafast perturbation mapping may be used for design optimization of photonic structures with reconfigurable functionalities.

[1] LP2N, CNRS-Institut d'Optique Graduate School-Univ. Bordeaux, 33400 Talence, France. [2] Physics and Astronomy, Faculty of Physical Sciences and Engineering, University of Southampton, Southampton SO17 1BJ, UK. [3] Optoelectronics Research Centre, University of Southampton, Southampton SO17 1BJ, UK. These authors contributed equally: Kevin Vynck, Nicholas J. Dinsdale. Correspondence and requests for materials should be addressed to K.V. (email: kevin.vynck@institutoptique.fr) or to O.L.M. (email: o.muskens@soton.ac.uk)

The industrialization of photonic integrated circuits as a high-volume, low-cost technology requires new approaches for optical testing[1]. Compared to the maturity of wafer-scale testing of nanoelectronics, available techniques for testing of optical chips are limited. Standard optical transmission measurements are able to address the transfer function of systems between all input and output ports[2, 3], however they do not allow access to the performance of individual elements in a complex circuit of many cascaded elements. As one of the possible solutions, introducing erasable gratings allows transmission characterization of certain parts of a device[4]. The most advanced tools for characterization of photonic devices rely on the use of a scanning perturbation placed in the near-field of the structure to infer information on the electromagnetic fields propagating within it[5]. Near-field optical techniques can provide very high resolution information on electromagnetic fields and are excellent research tools for detailed studies of new concepts. Their use in an industrial environment is less obvious due to the complexity of the nanoprobes and the requirement of near-field access.

Recently, a new technique, named ultrafast photomodulation spectroscopy (UPMS), that allows the remote optical characterization of photonic devices was introduced[6]. UPMS measures the impact of a local refractive index variation, created by ultrafast excitation of free carriers in the semiconductor material, on the transmittance between two ports of a photonic structure at a given frequency. The technique applies to arbitrary linear photonic systems, such as optical waveguides[7], multimode interference (MMI) devices[8], photonic-crystal structures[9], and subwavelength grating metamaterials[10]. By moving the perturbation position throughout the photonic structure, one obtains a transmittance perturbation map that is characteristic of the photonic structure, light excitation, and collection. Owing to its high operational speed and remote excitation, UPMS appears as a promising technology for non-contact wafer-scale testing of photonic chips. Developing the technique as a quantitative tool for optical testing requires establishing formally how the perturbation map stems from the light flow distribution in the photonic structure. A basic understanding of photomodulation maps may also open new possibilities for the design optimization of photonic devices with reconfigurable functionalities, obtained by exploiting multiple simultaneous ultrafast perturbations[11].

The sensitivity of electromagnetic fields to a perturbation has been studied earlier in the context of scanning near-field tips. In particular, it was shown that the fieldmap of a resonant mode can be recovered by analyzing the resonance spectral shift induced by a perturbation[12–16]. This approach is not restrictive to near-field investigations, as shown recently in a study exploiting a remote thermo-optic perturbation[17]. The problem considered here, however, differs in the fact that no spectral information is exploited, since the transmittance variation is measured at a given wavelength. A direct numerical resolution of the electromagnetic problem is also not a viable strategy since, besides not bringing much physical insight, one should repeat the simulations as many times as the number of perturbation positions. Therefore, it becomes a very computationally heavy task for large photonic devices.

Generally speaking, the study of how sensitive a response function is to design parameters variations is called a sensitivity analysis. Exploiting differential calculus, one can show that once the solution of the problem for the direct excitation is known, the variation of the response function with respect to any design parameter can be obtained with only one additional computation using an adjoint excitation[18]. This constitutes the basis of so-called adjoint methods, which have been used for sensitivity- and design optimization studies in many fields, including fluid dynamics[19], geophysics[20], microwave antennas[21–23],

phononics[24], and photonics[25–34]. An approach was recently proposed to predict the transmittance sensitivity between two ports of a nanophotonic device to the material permittivity[34]. Adjoint methods are however limited by the fact that they operate in a perturbative regime, valid for permittivity variations that weakly affect the response function of the system. In UPMS, the permittivity variations occur on the wavelength scale and can be quite large in amplitude, thereby resulting in substantial transmittance changes.

Here, we develop a general and rigorous method that allows the prediction of transmittance perturbation maps for arbitrary linear multiport photonic devices with great accuracy and minimal computational cost. By exploiting the Lorentz reciprocity theorem applied to normalized waveguide modes, we show that perturbation maps can be obtained simply by computing the response of the unperturbed system for two independently excited ports. A key step in the formalism is to accurately consider the local-field correction due to the presence of the perturbation. We implement highly-accurate corrections for small (dipole) cylindrical perturbations and propose approximate ones for large (wavelength-scale) perturbations. The full spatial perturbation maps of various optical multimode-interference devices are then measured experimentally by UPMS. An overall very good agreement with the predicted maps is found, thereby validating our theoretical method and experimental technique, as well as our physical understanding of perturbation maps in general. We show that the experimental access to perturbation maps using ultrafast photomodulation can be used to determine fabrication errors and tolerances in photonics manufacturing. We believe that the possibility to map the impact of local refractive index variations on mode transmittance could also open new perspectives in the framework of refractive-index engineering to enhance the functionalities of photonic components[11, 35–39].

## Results

**Theory and numerical results.** Let us consider an arbitrary photonic structure described by a relative permittivity tensor $\epsilon_b$ and coupled to one or several input and output waveguides, yielding a total of $N$ modes or ports, see Fig. 1. The $m$-th input mode, denoted as $\widetilde{\mathbf{\Phi}}_m^+$, is exciting the system and couples to the $n$-th output mode, denoted as $\widetilde{\mathbf{\Phi}}_n^-$, with a transmission coefficient $t_{mn}^0$. The $+$ (resp. $-$) superscript indicates ingoing (resp. outgoing) propagation.

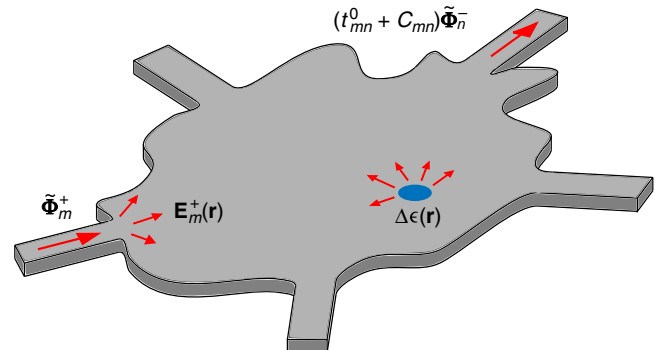

**Fig. 1** Illustration of the problem. An $m$-th ingoing waveguide mode $\widetilde{\mathbf{\Phi}}_m^+$ excites an arbitrary photonic system containing a perturbation $\Delta\epsilon$, yielding the total field $\mathbf{E}_m^+$. The transmission coefficient to the $n$-th outgoing mode $\widetilde{\mathbf{\Phi}}_n^-$ is $t_{mn} = t_{mn}^0 + C_{mn}$, where $t_{mn}^0$ is the transmission coefficient in the unperturbed system and $C_{mn}$ the coupling coefficient induced by scattering by the perturbation. The perturbation map is obtained by scanning the perturbation over the photonic system

A change of the relative permittivity to $\epsilon(\mathbf{r}) = \epsilon_b + \Delta\epsilon(\mathbf{r})$ in the system results in a change in the transmission coefficient due to light scattering by the perturbation as

$$t_{mn} = t_{mn}^0 + C_{mn}, \tag{1}$$

where $C_{mn}$ is the coupling coefficient between the $m$-th ingoing mode and the $n$-th outgoing mode via the scatterer $\Delta\epsilon(\mathbf{r})$. The relative variation of the transmittance, which can be measured experimentally, thus reads

$$\frac{\Delta T}{T} = \frac{|t_{mn}|^2}{|t_{mn}^0|^2} - 1 = \frac{|C_{mn}|^2}{|t_{mn}^0|^2} + 2\,\mathrm{Re}\left[\frac{C_{mn}}{t_{mn}^0}\right]. \tag{2}$$

In this section, we will derive analytical formulas for the coupling coefficient $C_{mn}$ as a function of the fields produced by exciting the unperturbed photonic system by the $m$-th and $n$-th ingoing modes, respectively, and provide an efficient method to estimate it for local perturbations placed at arbitrary positions.

To predict the impact of a perturbation on the transmission properties of a multi-port photonic device, we start from the vector wave equation for the total electric field $\mathbf{E}_m^+$ produced by an input mode $\hat{\mathbf{\Phi}}_m^+$ in the perturbed system, which, using the $\exp(-i\omega t)$ convention, reads

$$\nabla \times \nabla \times \mathbf{E}_m^+(\mathbf{r}) - k_0^2 \epsilon(\mathbf{r}) \mathbf{E}_m^+(\mathbf{r}) = i\omega\mu_0 \mathbf{J}_m^+(\mathbf{r}). \tag{3}$$

Here, $\epsilon$ is the relative permittivity tensor of the photonic structure, $\mathbf{J}_m^+$ is the current density source produced by $\hat{\mathbf{\Phi}}_m^+$ at the entrance of the photonic system and $k_0 = \omega/c$.

Decomposing the total field as the background and scattered field, $\mathbf{E}_m^+ = \mathbf{E}_{b,m}^+ + \mathbf{E}_{s,m}^+$, we reach the scattered field formulation

$$\nabla \times \nabla \times \mathbf{E}_{s,m}^+(\mathbf{r}) - k_0^2 \epsilon_b \mathbf{E}_{s,m}^+(\mathbf{r}) = k_0^2 \Delta\epsilon(\mathbf{r}) \mathbf{E}_m^+(\mathbf{r}). \tag{4}$$

The perturbation $\Delta\epsilon$ readily appears as a source for the scattered field in the unperturbed system with an associated current density

$$\mathbf{J}_{s,m}^+(\mathbf{r}) = -i\omega\epsilon_0 \Delta\epsilon(\mathbf{r}) \mathbf{E}_m^+(\mathbf{r}). \tag{5}$$

To calculate the coupling efficiency between the scattered field and an outgoing mode $\hat{\mathbf{\Phi}}_n^-$, we exploit the Lorentz reciprocity theorem in waveguide geometry[7], which, upon the sole assumption that the materials are reciprocal, i.e. $\epsilon^T = \epsilon$, where the superscript T denotes matrix transposition, establishes a relation between the current density sources and the fields produced in an arbitrary system for two different situations (typically, different excitations). Assuming that the modes of the input and output waveguides $\hat{\mathbf{\Phi}}$ form a complete set, meaning that an arbitrary propagating field in the waveguides can be written as a linear combination of waveguide (guided and radiating) modes, one finds that the excitation amplitude of an outgoing mode by a source is given by the overlap integral between the current density and the field produced at the source location by exciting the system with the reciprocal ingoing mode, that is

$$C_{mn} = -\tfrac{1}{4}\int \mathbf{J}_{s,m}^+(\mathbf{r}) \cdot \mathbf{E}_n^+(\mathbf{r})\mathrm{d}\mathbf{r}$$
$$= \frac{i\omega\epsilon_0}{4}\int \Delta\epsilon(\mathbf{r}) \mathbf{E}_m^+(\mathbf{r}) \cdot \mathbf{E}_n^+(\mathbf{r})\mathrm{d}\mathbf{r}, \tag{6}$$

where the 1/4 factor results from mode normalization with unitary power flux[40].

Equation (6) constitutes the basis of our approach. It is exact and applies to all multiple-input multiple-output photonic structures and all kinds of perturbations, provided that the materials involved are reciprocal. The downside of this generality is the fact that the fields in Eq. (6) are the total fields, which result from light scattering by the perturbation and are thus different for

each perturbation position, and not the background fields, which are the same for all perturbation positions. For spatially-localized perturbations, a transition ($T$) matrix may be computed and used to predict the total fields from the background fields[41]. As we will now show, simple analytical expressions for the coupling coefficient $C_{mn}$ can also be obtained in certain cases.

Hereafter, we will consider a planar dielectric waveguide system, such as a typical silicon-on-insulator (SOI) photonic device, in which light propagation is driven by the fundamental TE-mode. Since we are interested in planar wave propagation where out-of-plane losses are negligible, the problem can safely be treated within a 2D approximation, using the mode propagation constant to define an effective refractive index. We further assume that all materials have an isotropic (scalar) permittivity, $\epsilon = \epsilon\mathbf{I}$.

We start with the case of a small cylindrical perturbation, described by $\Delta\epsilon(\mathbf{r}) = \Delta\epsilon\Pi(|\mathbf{r} - \mathbf{r}_0|/2R)$, where $\Pi(\mathbf{r})$ is the Heaviside box function, $\mathbf{r}_0$ the position of the cylinder center and $R$ its radius. Formally, the induced electric dipole moment $\mathbf{p}$ of a perturbation is defined from the polarization $\mathbf{P}(\mathbf{r}) = \epsilon_0\Delta\epsilon\mathbf{E}(\mathbf{r})$ as $\mathbf{p} = \lim_{S_p \to 0}\int_{S_p}\mathbf{P}(\mathbf{r})\mathrm{d}\mathbf{r}$, where $S_p$ is the perturbation surface area. The dipole moment $\mathbf{p}$ is also related to the background field $\mathbf{E}_b$ via the scatterer polarizability $\alpha$ as $\mathbf{p} = \epsilon_0\epsilon_b\alpha\mathbf{E}_b(\mathbf{r}_0)$. In the limit of small perturbations ($\sqrt{\epsilon}R/\lambda \ll 1$), it is known from electrostatics that the total electric field in cylindrical and spherical cavities should be constant over the perturbation surface[42]. Equating the two expressions above for the dipole moment, we therefore reach the so-called local-field correction to the field

$$\mathbf{E}(\mathbf{r}_0) = \frac{\epsilon_b\alpha}{\Delta\epsilon S_p}\mathbf{E}_b(\mathbf{r}_0), \tag{7}$$

with $S_p = \pi R^2$. For cylindrical scatterers, the polarizability may be computed from the Mie scattering coefficient of order 1[43] ($a_1$ in TE-polarization) as $\alpha(\omega) = i8c^2a_1/(\omega^2\epsilon_b)$.

Considering again that $\mathbf{E}$ should be constant in small cylindrical perturbations, the integral in Eq. (6) can be simplified, and using Eq. (7), we eventually arrive to

$$C_{mn} = \frac{i\omega\epsilon_0\epsilon_b^2\alpha^2}{4\Delta\epsilon S_p}\mathbf{E}_{b,m}^+(\mathbf{r}_0) \cdot \mathbf{E}_{b,n}^+(\mathbf{r}_0). \tag{8}$$

The coupling coefficient $C_{mn}$ is now expressed only as a function of the background fields for two ingoing excitations. The relative transmittance variation $\Delta T/T$, obtained via Eq. (2), can thus be obtained for any perturbation position with only two electromagnetic computations for an unperturbed system and a scalar product.

To illustrate and validate our reciprocity-based method, we consider a typical $1 \times 2$ MMI device[8] operating at 1.55 μm. These systems have been investigated experimentally using UPMS in previous works[6, 11]. The simulations were made with a home-made implementation of the aperiodic Fourier-Model Method (a-FMM, see Methods)[44] and the input modes were normalized to have a unitary power. 

Figure 2a shows the norm of the electric fields, $\left|\mathbf{E}_{b,m}^+\right|$ and $\left|\mathbf{E}_{b,n}^+\right|$, created in the MMI without any perturbation for ingoing excitations by the fundamental modes of the input and upper output waveguides. For the excitation through the input waveguide, light is split in two, as expected, and coupled to the output waveguide fundamental mode with (modal) transmittance $T = 43.1\%$ for each. For the excitation through the output waveguide, a significant part of light is scattered out of the MMI, yet the transmittance between the two waveguide fundamental modes is identical to the above, due to reciprocity.

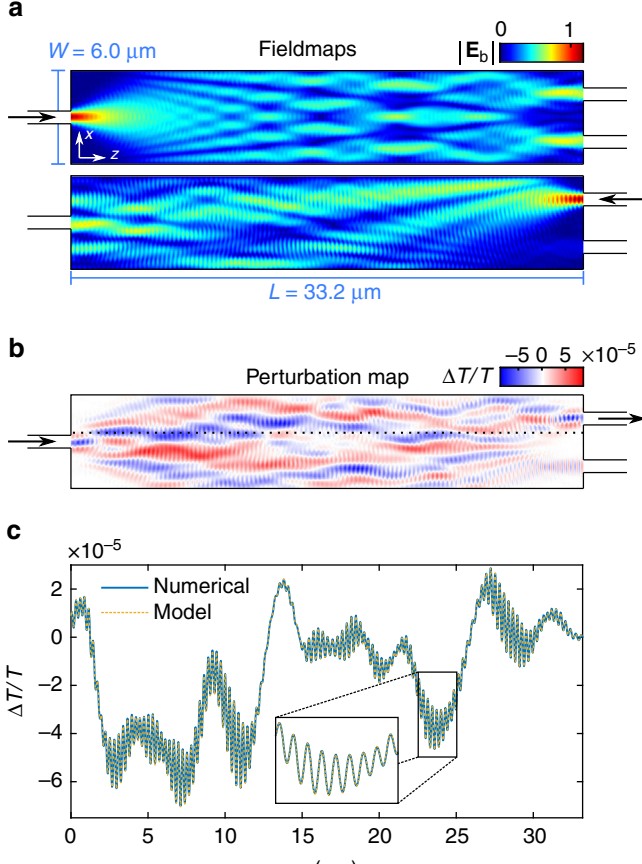

**Fig. 2** Validity of the reciprocity-based model for sub-wavelength perturbations. The photonic structure is a 1 × 2 MMI device of length $L = 33.2$ μm and width $W = 6.0$ μm operating at 1.55 μm (see Methods for more details). **a** Norms of the background electric field generated by ingoing excitations from the fundamental modes of the input and upper output waveguides, $\left|\mathbf{E}_{b,m}^{+}\right|$ and $\left|\mathbf{E}_{b,n}^{+}\right|$, respectively. **b** Resulting perturbation map as predicted using Eq. (8) for a cylindrical perturbation of diameter $2R = 10$ nm and refractive index variation $\Delta n = -0.25$. **c** Comparison between the perturbation curves for the same system along a cut (black dotted line in **b**) obtained from a-FMM performed for each perturbation position (fully-numerical, blue solid line), and predicted from the reciprocity-based model (yellow dotted line)

Having the two background fields, the perturbation map is then straightforwardly obtained from Eqs. (2) and (8), as shown in Fig. 2b in the illustrative case of a cylindrical perturbation of diameter $2R = 10$ nm and refractive index variation $\Delta n = -0.25$. The positive and negative values of $\Delta T/T$ indicate that the perturbation respectively increases and decreases the transmittance to the output waveguide mode. To verify that the prediction is correct, we then computed the transmittance variation $\Delta T/T$ due to the cylindrical perturbation by solving rigorously with a-FMM the electromagnetic problem for each perturbation position, thereby requiring as many calculations as the number of studied perturbation positions. The results of these simulations can be safely considered as exact. Figure 2c shows a quantitative agreement between our reciprocity-based model and the fully-numerical (point-by-point) computations. This clearly demonstrates the validity of our approach and its high accuracy in the dipolar approximation. As shown in the Supplementary Note 1, the local-field correction is essential to achieve quantitative agreement. Besides, let us emphasize that the gain in computational time is truly significant. To get the perturbation line along $z$

using a-FMM on a standard laptop computer, the point-by-point simulations required about 30–45 min while the predictions from our reciprocity-based model took less than 2 s.

In practical situations, such as in the UPMS experiments that will be presented in the next section, the perturbations occur on the wavelength scale and not on deep-subwavelength scales. As a result, they do not behave as electric dipoles (higher-order multipoles are also excited), such that the expressions derived in the previous section within the dipolar approximation cannot be applied.

As discussed above, Eq. (6) is valid for all perturbation sizes, however it requires calculating the total fields from the incident fields at each position of space, which may be inconvenient. Here, we propose to circumvent this issue by modeling large perturbations as simple 1D Fabry-Perot cavities. Optical systems such as MMIs are indeed essentially operating in a paraxial propagation regime, i.e., the variations of the field in the transverse direction $x$ are much slower than those along the propagation direction $z$. We may then neglect the transverse dimension. This model is approximate but, besides providing an analytical formula for $C_{mn}$ as a function of the background fields, as we will see, it has the important benefit to highlight the physical mechanisms underlying perturbation maps with large perturbations.

Let us then consider a 1D cavity of size $d$ centered on $z_0$ with refractive index $n_b + \Delta n$ in a background medium with refractive index $n_b = \sqrt{\epsilon_b}$. Following a classical derivation for layered media[45], it can be shown that the total fields in the cavity for light at normal incidence on the cavity, propagating along the $z$-axis, are related to the background fields as

$$\mathbf{E}_m^+(\mathbf{r}) = \frac{t_{12} \exp[i\omega\Delta n(z - (z_0 - d/2))/c]}{1 - r_{21}^2 \exp[i\omega(n_b + \Delta n)d/c]} \mathbf{E}_{b,m}^+(\mathbf{r}), \quad (9)$$

and

$$\mathbf{E}_n^+(\mathbf{r}) = \frac{t_{12} \exp[-i\omega\Delta n(z - (z_0 + d/2))/c]}{1 - r_{21}^2 \exp[i\omega(n_b + \Delta n)d/c]} \mathbf{E}_{b,n}^+(\mathbf{r}), \quad (10)$$

where $t_{12} = 2n_b/(2n_b + \Delta n)$ and $r_{21} = \Delta n/(2n_b + \Delta n)$ are the transmission and reflection amplitude coefficients at an interface between the background medium and the cavity, respectively. Inserting Eqs. (9) and (10) in Eq. (6), we therefore reach

$$C_{mn} = \frac{i\omega\epsilon_0}{4} \frac{t_{12}^2 \exp[i\omega\Delta nd/c]}{(1 - r_{21}^2 \exp[i\omega(n_b + \Delta n)d/c])^2} \Delta\epsilon \int_{S_p} \mathbf{E}_{b,m}^+(\mathbf{r}) \cdot \mathbf{E}_{b,n}^+(\mathbf{r}) d\mathbf{r}. \quad (11)$$

Several observations are in place. First, compared to the expression obtained in the dipolar approximation (Eq. (8)), calculating $C_{mn}$ requires performing an overlap integral over the perturbation surface. The wavelength-scale oscillations observed in Fig. 2b will therefore be smoothed out. Second, it is of utmost importance to note the presence of a dephasing term, $\exp[i\omega\Delta nd/c]$, which plays a leading role on the perturbation map. Indeed, it is the only part of the prefactor that survives for low-index-contrast perturbations, $\Delta n/n_b \ll 1$, as Eq. (11) simply reduces to

$$C_{mn} = \frac{i\omega\epsilon_0}{4} \exp\left[i\frac{\omega}{c}\Delta nd\right] \Delta\epsilon \int_{S_p} \mathbf{E}_{b,m}^+(\mathbf{r}) \cdot \mathbf{E}_{b,n}^+(\mathbf{r}) d\mathbf{r}. \quad (12)$$

Based on these considerations, let us finally discuss the applicability of this approximate model for UPMS experiments. The actual perturbation in the experiments does not exhibit a sharp refractive-index-change as in a cavity but rather a smooth Gaussian-like profile. This, in turn, should give a weak reflection coefficient, such that, as above, the leading mechanisms

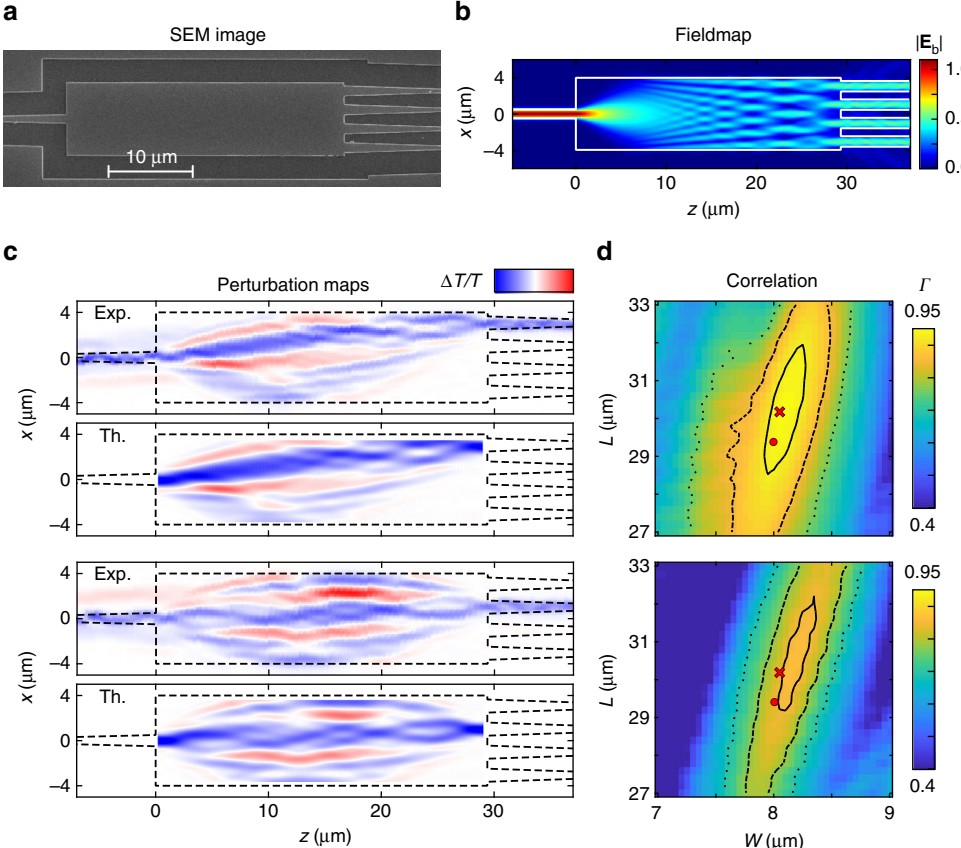

**Fig. 3** Experimental investigation of a 1 × 4 MMI and comparison with predictions. **a** SEM image of the MMI. The design parameters are $L = 29.4$ μm and $W = 8.0$ μm but the SEM image shows fabricated length $L = 30.2$ μm and width $W = 8.05$ μm. **b** Calculated fieldmap of the design device for ingoing excitation by the fundamental mode of the input waveguide. **c** Experimental and theoretical perturbation maps for the transmittance between the input waveguide fundamental mode and the top and second top waveguides fundamental modes, showing a remarkable agreement in their details. The amplitude of the color bar is ±0.25 for the experimental maps and ±0.35 for the theoretical ones. **d** 2D maps of the correlation $\Gamma$ for varying theoretical MMI region dimensions $L$ and $W$, for the two considered output modes. The red dot and cross indicate the MMI design and SEM measured dimensions, respectively, and the contour lines indicate a 2% (solid), 10% (dash), and 20% (dot) decrease of the correlation with respect to the maximum correlation

underlying the perturbation maps should be (i) the spatial averaging and (ii) the dephasing induced over the perturbation surface. We thus believe that the knowledge of the average size and refractive index of the perturbation are sufficient to predict the experimental perturbation maps with good quantitative agreement. This hypothesis will be verified in the next section, where our predictions will be compared directly to experimental results.

**Experimental perturbation maps and comparison with theory.** Experimental perturbation maps were obtained using UPMS (see Methods). In this technique, an 1550 nm infrared probe travels inside the photonic chip, while a synchronized 417 nm pump pulse is focused onto the surface of the silicon-on-insulator (SOI) waveguide[6]. The femtosecond optical pump pulse locally reduces the refractive index of the silicon via the plasma dispersion effect[46]. Due to the ultrafast nature of the pump, free-carrier concentrations far exceeding those achievable by electrical effects can be obtained[47]. The local perturbation in the refractive index modulates the transmittance of the device. By precisely recording the change in transmittance $\Delta T/T$ as a function of the perturbation position, it is possible to build up a 2D spatial map of the photomodulation response, providing a direct visualization of the impact of local refractive index variations on transmittance.

As an illustration of the capabilities of the technique, we first investigate a single-input quadruple-output (1 × 4) MMI as shown in the scanning electron microscopy (SEM) image of Fig. 3a. The MMI is designed to produce an equal split of intensity coupled to the four output ports. The corresponding multimode interference pattern in the device is shown via the calculated fieldmap in Fig. 3b. This local field pattern would be obtained using conventional scanning near-field optical microscopy. As discussed above, perturbation maps show the sensitivity of a specific output port transmittance, $\Delta T$, to a local perturbation. The perturbation maps of the differential transmission, $\Delta T/T$, obtained when detecting a specific output transmittance while scanning the pump spot over the device, are shown in Fig. 3c for the first and second output waveguide modes (the remaining two are given in the Supplementary Note 2). We obtain detailed maps showing structures on the length scale of the pump spot size of 740 nm. The experimental maps can be directly compared with predictions using Eq. (11). The theoretical maps were calculated for design parameters $L = 29.4$ μm and $W = 8.0$ μm, and experimentally determined values of $d = 740$ nm and $\Delta n = -0.25$ (see details in the Methods section). The amplitudes of the experimental and theoretical maps shown in Fig. 3c differ by about 0.1, which is likely to be due to the approximate values taken to reproduce the experiments. However, the experimental and theoretical maps show a remarkable level of agreement in their details and in relative values. This is a clear indication that

our approximate theoretical model and our experimental approach are fully consistent with each other in obtaining the perturbation map of photonic devices.

In order to quantify the similarity between the experimental and calculated perturbation maps, we calculated their two-dimensional correlation coefficient $\Gamma$. For this parameter, an area of equal size as the MMI region was scanned over the experimental map, and the correlation coefficient was calculated for each possible position, thus accounting for small relative shifts in the experimental alignment. We evaluated a library of theoretical perturbation maps scanning a wide range of design parameters and obtained the maximum correlation values for each calculated map in the library with the experimental result. Figure 3d shows 2D maps of the maximum correlation values for MMI widths $W$ and lengths $L$ around the fabricated device dimensions (red cross). As expected from MMI self-imaging theory, the device is significantly more sensitive to the width than to the length of the MMI region[8]. This is shown by the contours, which indicate 2, 10, and 20% decrease of the correlation with respect to its maximum (as high as 0.9448), the largest correlation coefficient clearly occurs around the dimensions observed by SEM ($L = 30.2\,\mu m$ and $W = 8.05\,\mu m$). The highest correlation values predict a slightly wider device, as shown by the 2% contour, which may be due to fabrication imperfections other than systematic deviations in $L$ and $W$.

While the above example illustrates the performance of a multiple-output device using a single input, the ultimate application of perturbation maps includes multiple-input multiple-output devices. Figure 4 investigates the response of a $3 \times 3$ symmetric MMI, which has been designed to couple each of the 3 inputs individually into 3 outputs at equal splitting ratios of around 1:3. This design requirement of equal splitting ratios can be fulfilled for a device of length $L = 88.0\,\mu m$ and width $W = 6.0\,\mu m$, as seen in the fieldmaps in Fig. 4b. This MMI length is more than twice as long as that previously studied, and this type of device is therefore much more sensitive to small fabrication imperfections. Figure 4c shows four out of the nine possible input-output configurations (the remaining five are given in the Supplementary Note 2), demonstrating the capability of the technique in providing a complete investigation of device performance. It is worth noting that the excitation/collection symmetry makes some of the perturbation maps very symmetrical. The symmetry in $x$ can be understood from the symmetry of the input and output modes, while the symmetry in $z$ stems from reciprocity (Eq. (6)), which predicts that the maps should look the same when flipping the input and output ports (see an experimental verification on a $1 \times 2$ MMI in the Supplementary Note 3). Contrary to the previous MMI, however, significant deviations are seen between the experimental maps and theoretical maps predicted for the design parameters. For instance, some of the experimental maps are not strictly symmetric, as they should be. These deviations indicate that the fabricated structure does not operate according to its ideal design. This appears clearly by calculating the correlation between experimental and theoretical maps in the 2D parameter space of $L$ and $W$, see Fig. 4d, which shows that the maximum correlation is obtained for a width reduction by ∼0.1 μm, as confirmed by the SEM image. As shown in the Supplementary Note 4, better agreement between the experimental and theoretical perturbation maps is indeed obtained when using the device parameters extracted from SEM images for the prediction.

**Field intensity recovery from perturbation maps**. The perturbation maps reflect the spatially-distributed contribution of a localized perturbation in the device to the transmittance to a particular output. Areas of negative $\Delta T/T$ indicate reduced transmittance, with light being diverted away from the considered output mode, while positive $\Delta T/T$ areas indicate that the perturbation redirects part of the light towards the output mode. This light could have originally been either coupled to a different output or lost through scattering out of the device[11]. Consider now the case where light coupled into the MMI from a specific input mode couples with nearly unitary transmission to a single output mode. In this case, the perturbation maps can exhibit only negative $\Delta T/T$ areas. In such a situation, intuitively, one may expect the extinction due to the perturbation to be proportional to the field intensity. This would imply that perturbation maps could provide a direct visualization of the light intensity in the photonic device.

Formally, one can show from Lorentz reciprocity that, when $\left|t_{mn}^0\right| = 1$, the field generated from an ingoing mode in the output waveguide $\mathbf{E}_{b,n}^+$ should be equal to the complex conjugate of the field generated from an ingoing mode in the input waveguide, $\left(\mathbf{E}_{b,m}^+\right)^\star$. This is equivalent to considering the time-reversed solution of the direct problem. Under this condition, the coupling coefficient $C_{mn}$—in Eq. (8) for small cylindrical perturbations or Eqs. (11) and (12) for large low-index perturbations - becomes directly proportional to the field intensity $\left|\mathbf{E}_{b,m}^+\right|^2$. Assuming then that the transmission variation is small compared to the transmission through the unperturbed system, $C_{mn} \ll t_{mn}^0$, one indeed finds via Eq. (2) that $\Delta T/T$ should be directly proportional to the field intensity (see the formal derivation in the Supplementary Note 5).

We have tested this possibility to recover the field intensity from perturbation maps with the case of a $1 \times 2$ MMI with a silica cladding. This cladding would prevent any recovery of the intensity from near-field measurements. Instead of considering the transmittance to the fundamental modes of the individual output waveguides, we studied the fundamental mode of the pair of output waveguides, which yields a transmittance of about 87%.

Figure 5a shows the intensity distribution $\left|\mathbf{E}_{b,m}^+\right|^2$ for excitation by the fundamental mode of the input waveguide and the upper panel in Fig. 5b the perturbation map $\Delta T/T$ predicted from our model for large perturbations (Eq. (11)) between the fundamental input and output modes. The two maps clearly share the same features, the differences being explained by the imperfect transmission and the perturbative limit $C_{mn} \ll t_{mn}^0$ which is not completely fulfilled. Note also that the finite perturbation size causes a reduction of the finer details of the intensity. Experimentally, we measured the total output transmission of the $1 \times 2$ MMI by combining the two outputs using a reversed $2 \times 1$ MMI. Results are shown in the lower panel of Fig. 5b, where an excellent agreement with theory is found. Quite remarkably, we were able to resolve as many as five intermediate self-imaging points.

## Discussion

To summarize, we developed a rigorous theoretical understanding of photomodulation maps, making UPMS a quantitative technique to characterize photonic devices. We believe that our theoretical and experimental study contributes to the established field of photonic probing techniques in several ways.

First, we developed a general and rigorous theoretical method to predict the transmittance perturbation map of arbitrary linear photonic systems due to local refractive index perturbations. In line with adjoint methods, only two electromagnetic computations are required to obtain the full transmittance perturbation

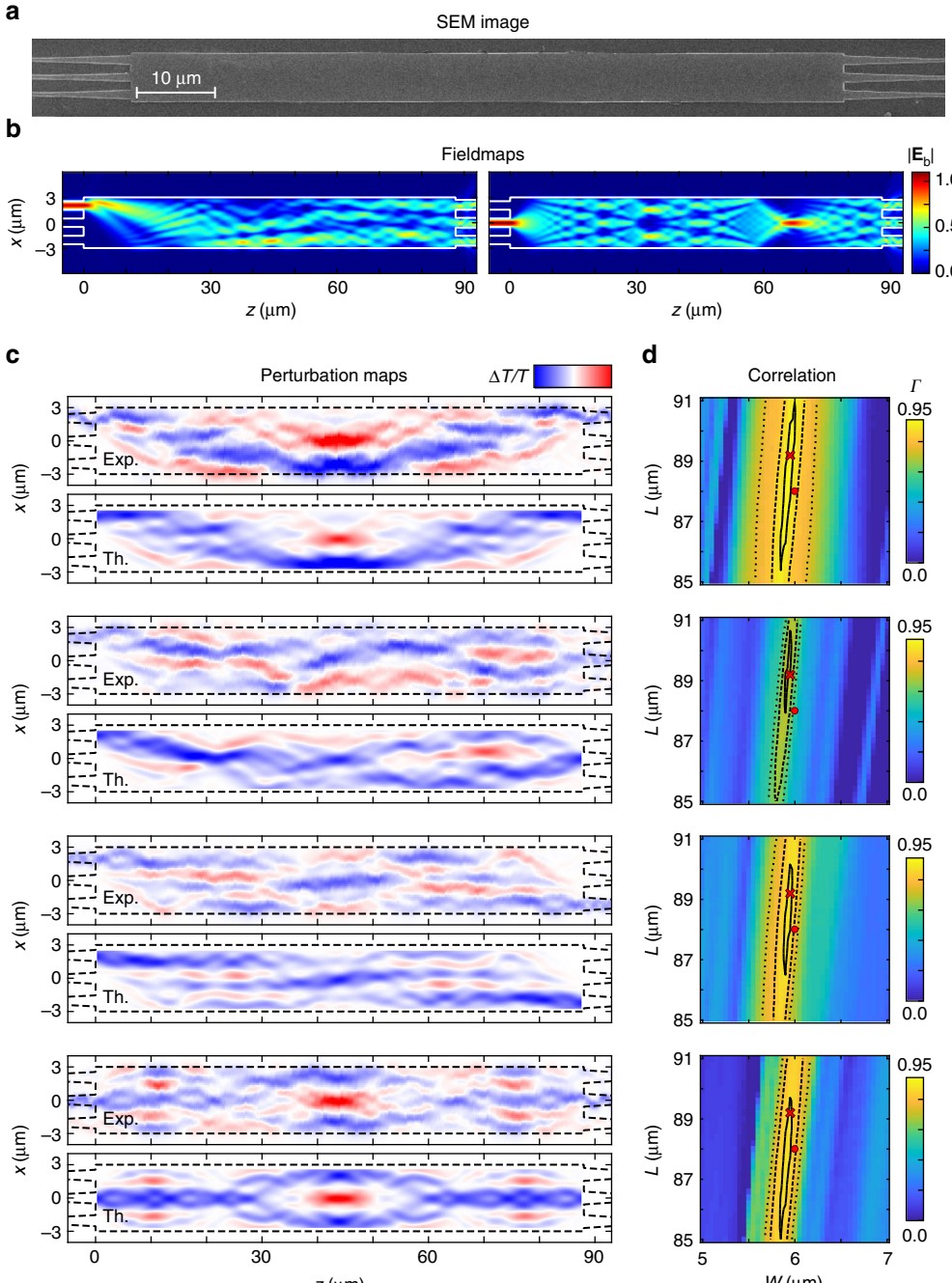

**Fig. 4** Experimental investigation of a 3 × 3 MMI and comparison with predictions. **a–d** Same as in Fig. 3 for a MMI with designed length $L = 88.0$ μm and width $W = 6.0$ μm, and SEM measured length $L = 89.2$ μm and width $W = 5.95$ μm. The scale has been halved in the propagation axis for **b**, **c**. The amplitude of the color bar in **c** is ±0.3 for the experimental maps and ±0.4 for the theoretical ones

map. Contrary to most adjoint-based formalisms, however, ours does not rely on any specific discretization scheme, making it applicable to any electromagnetic simulation solver. The method is also virtually exact and not restricted to small (in size and permittivity change) perturbations, provided that the local-field corrections induced by the perturbation are properly handled. While these corrections may be computed numerically via a $T$-matrix, we derived exact and approximate analytical expressions for small cylindrical perturbations and large low-index-contrast perturbations, respectively. The latter are necessary for reproducing experimental results obtained using UPMS. We believe that our theoretical method used with an efficient

electromagnetic solver such as a-FMM constitutes a new powerful analysis approach of scanning perturbation experiments. It would be interesting in a future work to generalize and apply the method to nanostructured photonic components with superior performances, such as sub-wavelength grating MMIs that exhibit an anisotropic effective permittivity[39]. Future studies may also consider using the method for design optimization of multi-port photonic devices, as ensembles of such perturbations using UPMS have been shown to enable reconfigurable photonic systems[11].

Second, we performed UPMS experiments to measure the perturbation maps of various MMI devices. A very good overall agreement was found with the perturbation maps predicted by our

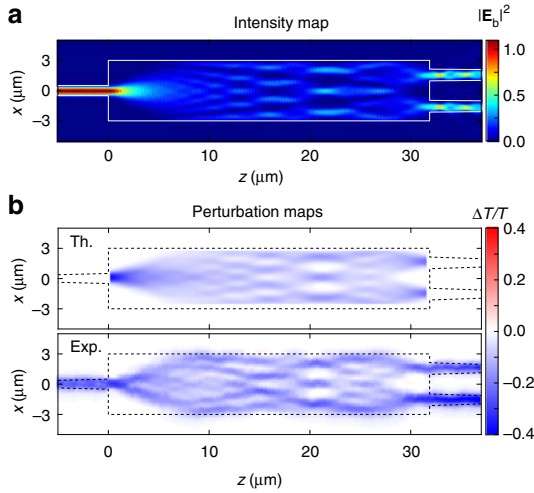

**Fig. 5** Field intensity recovery for a 1 × 2 MMI. We considered here a MMI device with length $L = 31.875\,\mu m$ and width $W = 6.0\,\mu m$, and covered by a silica cladding. **a** Simulated field intensity map for excitation by the input waveguide fundamental mode. **b** Perturbation maps predicted with our model for the coupling between the fundamental mode of the input waveguide and the fundamental mode of the pair of output waveguides, and measured experimentally by combining the two MMI outputs using a reverse 2 × 1 MMI splitter

method, demonstrating our capabilities to predict and measure the perturbation maps of photonic devices. Interestingly, we observed that perturbation maps exhibit a high sensitivity to fabrication deviations from ideal design. These deviations were readily observed via a parametric correlation study between experimental and predicted maps. The value of this approach as a tool for optical testing is given by a combination of experimental accuracy and the reliability of the used model. At this stage, we have only taken into account the MMI width $W$ and length $L$ to define a 2D parameter space to categorize devices. Other parameters include thickness variations of the SOI wafer, variations in the cladding layer, refractive index variations including thermal variations over the chip, and fabrication imperfections. As the perturbation method relies on comparison with a model, a full analysis would need to include all of these aspects, some of which bear similar effects on the output performance. For example, variations in effective refractive index caused by variable SOI or cladding variations will result in different effective values of $L$ and $W$. Fabrication imperfections and gradients in device dimensions along the wafer can both cause asymmetry in the structure, which could explain some of the experimental results obtained in our work. At this stage, our technique cannot fully interpret these results, which would require a more detailed study. Nevertheless, the absolute correlation amplitude may be used as a validation tool to benchmark fabricated device performance against the ideal device design. We envision that, with further quantitative investigations, ultrafast perturbation maps could be deployed into industry as a tool for diagnostics of device runs without requiring additional structural information such as scanning electron microscopy. A vital factor to address towards this aim will be characterization speed. For typical scan parameters used here, we achieved scan speeds on the order of a few micrometers per second, resulting in full scanning times on the order of minutes. This could be significantly improved to the order of a few seconds per device through the use of a faster scanning system, such as a mirror galvanometer, combined with higher frequency optical modulators to reduce dwell times.

Finally, we showed using UPMS that transmittance perturbation map measurements could reveal the light intensity

distribution in high-transmittance devices. This possibility relies on the assumptions that the transmittance of the photonic system is nearly unity and that the transmission variation induced by the perturbation is very small. Nevertheless, we believe that much remains to be understood and tested, such as the capability of the approach to reveal light propagation in imperfect systems. Deeper investigations on this aspect may result in a paradigm change in imaging techniques. We therefore hope that our study will motivate further investigations and applications in photonics research and manufacturing.

## Methods

**Numerical modeling.** The simulations were performed using an in-house aperiodic Fourier Modal method (a-FMM)[44]. The a-FMM is a fully-vectorial method relying on Fourier expansion techniques and perfectly matched layers, which has been shown to provide numerical predictions with a high accuracy and a fast convergence[48]. It relies on a scattering-matrix formalism to describe accurately the coupling between modes propagating in the input waveguide, MMI, and output waveguides. Simulations were performed in 2D, at a wavelength of 1.55 μm, using effective indices 2.833 and 1.741 for the core and surrounding media to account for the vertical confinement of the fundamental TE-mode of a 220/100 nm thick silicon rib waveguide. For the 1 × 2 MMI investigated in Fig. 5, we used 2.8502 and 1.4446 as the core and surrounding effective material indices, since the structure consisted in silica-covered strip waveguides. In all systems, the input and output waveguides were 1 μm wide and the fundamental waveguide modes were considered as input and output. For the point-by-point a-FMM simulations of Fig. 2b, since the polarizability α in Eq. (8) was that of a cylinder, the 10 nm diameter cylindrical perturbation was discretized into about 80 smaller squares.

**Sample fabrication.** The MMI devices were fabricated from a 220 nm thick silicon layer on top of a 2 μm thick SiO₂ layer employing electron beam lithography and reactive ion etching. The MMI lengths and widths are given in the main text. Single mode 500 nm wide waveguides are tapered to 1 μm at the MMI boundaries, ensuring an adiabatic size conversion of the fundamental mode over a 10 μm distance.

**Ultrafast photomodulation spectroscopy.** The experimental setup is shown in Fig. 6. Compared to earlier studies[6], a new and improved setup was developed using a 200 fs mode-locked Ti:Sapphire laser operating at 80 MHz, in conjunction with an optical parametric oscillator. The second harmonic of the laser at 417 nm wavelength was used as the pump. Through the use of synchronized ultrafast pulses for both the pump and probe, the effect of the perturbation can be observed during the 200 fs window of the probe pulse, thereby drastically improving signal-to-noise ratio. The pump was focused onto the surface of the device using a 100× objective with a numerical aperture of 0.5, resulting in a perturbation spot size of 740 nm at the focus with a fluence of 60 pJ μm⁻². The same objective was used to image the device for alignment purposes. The scans were carried out for 1550 nm probe pulses, coupled into and out of the device via optical fibers and etched silicon gratings, and delayed by 3 ps compared to the pump pulses. Transmitted intensity was modulated using a mechanical chopper and was detected using an InGaAs photodetector and lock-in amplifier. The effective refractive index change of $\Delta n = -0.25$ was obtained experimentally as in the ref. [6] by observing the shifting of

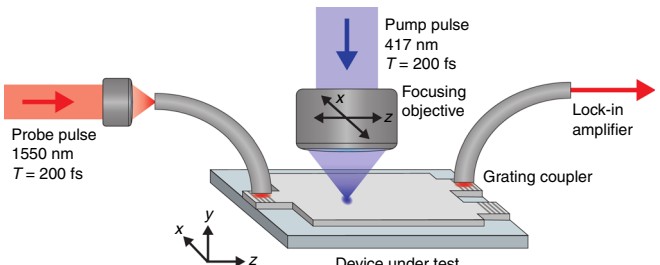

**Fig. 6** Illustration of the experimental setup. Infrared probe pulses are grating-coupled into the waveguide, travel through the device and are then out-coupled by another grating. The transmitted light is collected by an optical fiber and the transmission is recorded by an InGaAs photodetector connected to a lock-in amplifier. The pump pulses are incident perpendicular to the device's surface, and the delay time between the two pulses is controlled via a variable time delay stage. The pump focusing objective is mounted to a 3D piezo stage which allows the pump spot to be focused and scanned over the device

fringes in the transmission spectra of an asymmetric Mach-Zehnder interferometer (MZI) for a fixed perturbation length and vaying pump fluence on one of the waveguide arms. The imaginary part of the refractive index change was estimated theoretically from a Drude-Lorentz free-carrier model[46] to $Im(\Delta n) = 3 \times 10^{-4}$. It was found to have an insignificant impact on the theoretical predictions of perturbation maps and was thus neglected in the modeling.

**Data availability**. All data supporting this study are openly available from the University of Southampton repository at https://doi.org/10.5258/SOTON/D0485.

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

## Acknowledgements

This work was partly funded by the Royal Society via the project "Light shaping on a chip with nanophotonics and complexity". The authors acknowledge support from EPSRC through grant EP/J016918/1 and EP/L00044X/1. The SOI samples were fabricated in the frame of the EPSRC CORNERSTONE project (EP/L021129/1).

## Author contributions

K.V. developed the theory and the numerical codes. N.J.D. and K.V. performed the numerical simulations. R.B., A.Z.K., S.A.R., L.C., and D.J.T. fabricated the samples. N.J.D., B.C., and R.B. developed the UPMS setup. N.J.D. performed the perturbation map measurements. N.J.D., K.V., and O.L.M. analyzed the data. G.T.R. supervised the sample fabrication. P.L. supervised the theory development. O.L.M. designed the study and

supervised the whole project. K.V., N.J.D., and O.L.M. wrote the manuscript, with feedbacks from all co-authors.

## Additional information

**Competing interests:** The authors declare no competing interests.

