## [Peer Review File · Nature Communications]

Reviewers' comments:

Reviewer #1 (Remarks to the Author):

This paper by Vynck et al. reports on a rigorous theoretical analysis of ultra-fast photomodulation maps and its experimental implementation as a quantitative technique to characterize integrated photonic devices. The paper quality and depth are outstanding. The work unites in a balanced way a new theoretical framework with sound experimental validations and practical implementations. A rigorous theoretical analysis is brilliantly derived and explained, starting with a clear problem definition, followed by a rigorous model, an exact analytical solution for sub-wavelength excitation limit, and a practical approximate solution for a realistic perturbation spot size (pump laser spot size 740 nm). Experimental validation is solid and convincing. A very good agreement between theory and experiment is shown on several practical examples of multimode interference (MMI) devices (1x4, 3x3). It is also shown, by a parametric correlation study, that the proposed technique is promising for device tolerance and fabrication error analysis. I believe this work can potentially revolutionize diagnostics of integrated photonic devices, for both academic research and industry (wafer scale non-contact optical testing). I venture to say that this foundational work will likely greatly influence thinking in the field, and the paper will become a cornerstone. I strongly recommend the paper for publication in Nature Communications, with some minor revisions. Specifically, I recommend authors address the following points:

Page 5, 3rd line following Eq. 2, it should be "outgoing mode n " (not "ingoing mode n ").

Page 6, 1st paragraph: "the modes of the input and output waveguides form a complete set in the waveguides". I suggest authors remind the reader not necessarily expert in waveguide theory, what the notion of complete set of (normal) modes means.

Eq. 2 is of fundamental importance for the technique, which is based on monitoring transmittance relative variations. My feeling is that the technique can be even further improved if the phase information could be preserved in Eq. 2. The phase variations at different ports can be readily measured using coherent techniques. I think it would be very interesting if authors could briefly comment about this option. I do not suggest including the phase term in the manuscript now, just to consider this possibility and comment, in case authors think this could be of interest.

Page 2, 2nd paragraph, in "Recently, some of us introduced ultrafast ...", does not read quite well.

The authors mention interesting perspectives of their technique "for the design optimization of photonic components with enhanced functionalities" [10, 34-36]. I feel the chosen references [34-36] are not the most appropriate in this context. Just an example, one can argue that the key novelty in [35] is not enhanced functionality but new inverse design methodology. The functionality shown in [35] is actually rather limited, -11 dB channel isolation is insufficient for most practical applications. Furthermore, it is not clear how the present technique with the spot size of 740 nm could be used to map a device with a footprint as small as 2.8 mic x 2.8 mic [ref 35], I do not think this is plausible.

In this context, I would like to point out that many advanced components with unprecedented functionalities have been demonstrated based on the principle of subwavelength grating metamaterial index engineering. The following references would be more appropriate when talking about "photonic components with enhanced functionalities" (page 4, 2nd paragraph, last sentence):

"Refractive index engineering with subwavelength gratings implemented in a highly efficient microphotonic coupler and a planar waveguide multiplexer," Optics Letters 35, 2526 – 2528

(2010).

“Waveguide sub-wavelength structures: a review of principles and applications,” *Laser and Photonics Reviews*, 9, 25-49, 2015 (review article).

Also, it would be very interesting if authors could specifically comment about potentially using their technique for design optimization and/or characterization of advanced MMI devices with enhanced functionality as reported in:

“Ultra-broadband nanophotonic beamsplitter using an anisotropic sub-wavelength metamaterial,” *Laser Photonics Rev.* 10, No. 6, 1039–1046 (2016) / DOI 10.1002/lpor.201600213.

Reviewer #2 (Remarks to the Author):

In this manuscript a very efficient numerical method is introduced in order to carry out the analysis of the perturbation maps measured through the Ultrafast Photomodulation Spectroscopy. The main goal of the experimental method introduced in ref. 6 is to provide a tool for probing photonic integrated circuits which is suitable for industrial environment. In this respect, the method is way more accessible than scanning near field techniques, requiring probes. On the other hand, it is true that subwavelength resolution is not necessary in this context.

The authors make the point that the connection between the experimental perturbation maps and field inside the circuits requires heavy numerical computation.

Hence, a technique based on the Lorentz reciprocity theorem and assuming linear perturbations is used in order to reduced the computation burden drastically.

It is first shown that using a suitable local field correction, an excellent agreement is found between a full numerical solution and the prediction based on the sole calculation of the background fields.

Furthermore, a Fabry-Perot model is introduced to account for the perturbations corresponding to UPMS. It is shown that the prediction of this method is much closer to experiments where coherent effects appear to be averaged out. This very efficient computation method allows to perform a sensitivity analysis through the correlation of the experimental and numerical maps and therefore to infer the actual geometry of the device. Finally, it is shown that the method can be used to retrieve the actual field inside the device.

The manuscript is very well written, clear, and all the claims are well substantiated. The technical level is high and the proof are given with rigor and clarity. I could not find any issue with the text and formulas and the figures are very clear.

Therefore, the manuscript is well within the criteria of quality for Nature journals.
I recommend publication

Reviewer #3 (Remarks to the Author):

In the work “Ultrafast perturbation maps as a quantitative tool for testing of multi-port photonic devices” by K. Vynck et al., the authors describe a methodology for characterizing multi-port photonic devices using a local perturbation technique where scatters within the body of the device result in intensity modulations at an output port of interest. This results in a finger-print that can be used to compare such perturbation maps of experimentally tested devices to theoretically modeled devices to get a relative sense for how closely an as-fabricated device performs relative to an ideal design. Furthermore, in special cases (where the device-under-test represents a unitary operator mapping the input modes to output modes) these maps can be interpreted as measurements proportional to the actual intensity profile when all the output ports are taken into consideration simultaneously.

Towards this end, the authors describe a method for simplifying the theoretical analysis through the use of a reciprocity based construction (which is shown to be substantially faster than a FMM simulation for example), and experimentally validate the expected perturbation maps showing good agreement.

The reviewer finds this study novel, well written, and of interest to a broad audience including those interested in experimental characterization of passive devices, those interested in perturbation theory in guided-wave optics, and those in a commercial setting looking for characterization solutions during high-volume manufacturing and test. In general, this paper is recommended for publication by the reviewer, and the following comments are meant to help the authors improve the quality of the manuscript.

- It is clear that there are advantages to using a pulse for the pump, since higher optical intensities can result in higher local index perturbations from direct absorption. It would seem that for the probe, a CW input would be the most valid since it represents a \sim single frequency for the interference-based device-under-test. What was the motivation for also using a pulse for the probe?
- In addition to free-carrier-dispersion introduced by the pump, which provides the scattering perturbation used in the analysis, the presence of carriers introduces free-carrier-absorption, which will also act to suppress photons from the output port under test. It would be interesting if the authors showed what the expected change to the imaginary part of the index was from the incident pump, and if this effect can bring the measured vs modeled perturbation maps closer to agreement, or if the absorption is small enough to be safely neglected.
- The authors describe the index perturbation of -0.25 as experimentally determined in the experimental results section. It would be interesting if the authors included how this experimentally determined number was measured, in the supplementary information section.
- The authors show that the measured perturbation maps are correlated most strongly for a different waveguide geometry as intended in the fabrication, suggesting that the delta is from fabrication variation which is confirmed with SEM. It would be very interesting if the authors plotted, in addition to the measured vs. modeled perturbation map, a measured vs. modeled vs modeled using the MMI parameters that have the highest correlation. This would especially be interesting in cases where there is most disagreement, such as the 3x3 MMI case outlined in Fig.4.
- One of the motivations for this type of experimental analysis is characterization of optical components at the wafer-level in a manufacturing environment, as motivated by the authors in

the introduction. Such a characterization would take time, which is a vital factor in enabling wafer-level deployment. It would be interesting if the authors described how much time a single device characterization consumed using their setup, and a rough estimate on how fast a commercial system could test a single device, to help the reader understand the efficacy of this technique in a high-volume manufacturing.

Reviewed by: Dr. Jeffrey B. Driscoll

We would like to thank the reviewers for the thorough examination of our manuscript, detailed comments, and suggestions of improvements. All points have been addressed. The modifications made in the manuscript are highlighted in blue. We believe that they further improve the quality of the paper. We hope that the revised version will be fully satisfactory.

Point-by-point response to all comments/questions are provided below.

REPLY TO REVIEWER #1

Page 5, 3rd line following Eq. 2, it should be “outgoing mode n ” (not “ingoing mode n ”).

The coupling coefficient C_{mn} describes the coupling between an ingoing mode m and an outgoing mode n , yet we evaluate it from the fields generated by exciting the system by the ingoing modes m and n .

Action taken: We rewrote the text above and below Eq. (2) to avoid any possible confusion.

Page 6, 1st paragraph: “the modes of the input and output waveguides form a complete set in the waveguides”. I suggest authors remind the reader not necessarily expert in waveguide theory, what the notion of complete set of (normal) modes means.

We agree that defining completeness would be useful to readers. A modal basis is complete if an arbitrary field can be approximated arbitrarily well by a linear combination of such modes. In waveguides, radiating and guided modes form a complete basis for propagating fields.

Action taken: We completed the sentence to explain what the notion of a complete set of modes means.

Eq. 2 is of fundamental importance for the technique, which is based on monitoring transmittance relative variations. My feeling is that the technique can be even further improved if the phase information could be preserved in Eq. 2. The phase variations at different ports can be readily measured using coherent techniques. I think it would be very interesting if authors could briefly comment about this option. I do not suggest including the phase term in the manuscript now, just to consider this possibility and comment, in case authors think this could be of interest.

We agree, this is an interesting suggestion. Phase variations at different ports may indeed be measured with interferometric measurements. We have tested this possibility by computing numerically the maps of the real and imaginary parts of (C_{mn}/t_{mn}^0) for the realistic perturbations of 740 nm in size in the case of a 1x2 MMI for coupling between the fundamental modes of one input and one output waveguide. The results are given in the figure below.

As expected from Eq. (2), the real part of (C_{mn}/t_{mn}^0) closely resembles the perturbation map $\Delta T/T = |C_{mn}/t_{mn}^0|^2 + 2\text{Re}(C_{mn}/t_{mn}^0)$ (see Fig. S2) since this is the leading term for $C_{mn} \ll t_{mn}^0$. The imaginary part of (C_{mn}/t_{mn}^0) has a quite similar behavior. As C_{mn} is defined from the overlap integral of the fields (see Eq. (6)), the spatial resolution is necessarily limited by the perturbation size for both (C_{mn}/t_{mn}^0) and $\Delta T/T$. Phase measurements would not lever this limitation, at least.

Action taken: Being unsure about the real potential improvement of the technique by measuring phase variations (due to the overlap integral made at the level of C_{mn}), we prefer not to add a comment on this aspect in this work.

Page 2, 2nd paragraph, in “Recently, some of us introduced ultrafast ...”, does not read quite well.

Action taken: We modified the sentence.

The authors mention interesting perspectives of their technique “for the design optimization of photonic components with enhanced functionalities” [10, 34-36]. I feel the chosen references [34-36] are not the most appropriate in this context. Just an example, one can argue that the key novelty in [35] is not enhanced functionality but new inverse design methodology. The functionality shown in [35] is actually rather limited, -11 dB channel isolation is insufficient for most practical applications. Furthermore, it is not clear how the present technique with the spot size of 740 nm could be used to map a device with a footprint as small as 2.8 mic x 2.8 mic [ref 35], I do not think this is plausible.

We completely agree with the comments of the reviewer. We actually did not intend to make the reader believe that our technique could be used specifically to create new functionalities on such small-footprint photonic devices. Sorry for the confusion. The references were given as examples from the literature where the refractive index is engineered to improve or create new functionalities in photonic devices.

Action taken: This sentence has been modified to clarify that the references were given as general examples of refractive-index engineering on photonic structures. We also added the references suggested by the reviewer (see below) as they are very relevant in this context.

In this context, I would like to point out that many advanced components with unprecedented functionalities have been demonstrated based on the principle of subwavelength grating metamaterial index engineering. The following references would be more appropriate when talking about “photonic components with enhanced functionalities” (page 4, 2nd paragraph, last sentence):

“Refractive index engineering with subwavelength gratings implemented in a highly efficient microphotonic coupler and a planar waveguide multiplexer,” Optics Letters 35, 2526 – 2528 (2010).
“Waveguide sub-wavelength structures: a review of principles and applications,” Laser and Photonics Reviews, 9, 25-49, 2015 (review article).

These works are very interesting and very relevant to our work. We are pleased to cite them.

Action taken: The review article is now cited in the second paragraph, as an example of photonic system for which the photomodulation technique applies, alongside with optical waveguides, MMIs and photonic crystal structures. The other two references (including the one below) are now cited at the end of the introduction, as examples of works using “refractive-index engineering to enhance the functionality of photonic components”.

Also, it would be very interesting if authors could specifically comment about potentially using their technique for design optimization and/or characterization of advanced MMI devices with enhanced functionality as reported in:

“Ultra-broadband nanophotonic beamsplitter using an anisotropic sub-wavelength metamaterial,” Laser Photonics Rev. 10, No. 6, 1039–1046 (2016) / DOI 10.1002/lpor.201600213.

In the paper suggested by the reviewer, the authors consider a MMI made of a sub-wavelength grating structure that yields an effective permittivity that is anisotropic. The anisotropy allows achieving unprecedented functionalities. Experimentally, UPMS can certainly be used to characterize such structures. Theoretically predicting the perturbation maps of such devices however appears as an important challenge. We believe that it would be extremely interesting in a future work to generalize and apply the theory developed in our work to nanostructured media. The difficulty will be to correctly handle the local-field corrections induced by a permittivity variation in such a medium. A starting point may be to tackle the problem in the effective-medium sense to allow us following the same steps as reported in this work, yet for an anisotropic permittivity (here, we assumed that it was a scalar).

Action taken: As explained above, the review paper on subwavelength grating structures suggested by the reviewer is now cited in the introduction to emphasize the fact that UPMS can be used on such structures. We added a sentence in the discussion to motivate future theoretical efforts generalizing our method to nanostructured media, such as the anisotropic sub-wavelength grating structures cited by the reviewer.

REPLY TO REVIEWER #2

No correction requested.

REPLY TO REVIEWER #3

It is clear that there are advantages to using a pulse for the pump, since higher optical intensities can result in higher local index perturbations from direct absorption. It would seem that for the probe, a CW input would be the most valid since it represents a ~ single frequency for the interference-based device-under-test. What was the motivation for also using a pulse for the probe?

The motivation for the use of a pulse for the probe was indeed missing in our manuscript. We used

an optical pulse duration of 200 fs at a laser repetition rate of 80 MHz. Accounting for the free-carrier lifetime, this means that the device is perturbed about 1% of the time. As a result, the measured effect of the pump on transmission would be substantially reduced over the duration of the integration time of our lock-in amplifier, thereby decreasing our signal-to-noise ratio. By using synchronized ultrafast pulses, we are able to observe the effect of the pump only during the 200 fs window of our probe pulse, thereby drastically improving the signal-to-noise ratio. Considering also that the bandwidth of the MMI device is larger than the bandwidth of the probe pulse, we conclude that our approach approximates very well the result that would be obtained with a CW probe (as in the simulations).

Action taken: We added a sentence on the motivation for using a pulse for the probe in the Methods section.

In addition to free-carrier-dispersion introduced by the pump, which provides the scattering perturbation used in the analysis, the presence of carriers introduces free-carrier-absorption, which will also act to suppress photons from the output port under test. It would be interesting if the authors showed what the expected change to the imaginary part of the index was from the incident pump, and if this effect can bring the measured vs modeled perturbation maps closer to agreement, or if the absorption is small enough to be safely neglected.

This is an interesting comment, which was also not explicitly discussed in the original version of the manuscript. The pulsed pump indeed introduces absorption in the material, which should translate into an imaginary part in the refractive index variation Δn .

To answer this point, we computed numerically the correlation between the measured perturbation map and predicted perturbation maps with varying absorption coefficients in the case of the 3x3 MMI (the one that deviates more from theory) using the MMI parameters extracted from SEM. Results are given in the figure below for two input/output configurations (top-to-top and middle-to-top mode coupling). The inset shows a zoom of the curves for weak absorption.

It appears very clearly that including absorption deteriorates the agreement between theory and experiments. In fact, the absorption can safely be neglected in our case. Theory predicts, using the Drude-Lorentz free-carrier equations (Ref. [46] by Soref and Bennett), that the imaginary part of the refractive index change in our case (fluence of 60 pJ/ μm^2) should be $\text{Im}(\Delta n) = 3.10^{-4}$. The

correlation therefore goes from 0.9322 to 0.9319 and from 0.7156 to 0.7149. Considering the other approximations made (exact size of the perturbation, validity of the Fabry-Perot expression for large perturbations), this variation is clearly irrelevant.

Action taken: We added a sentence in the Methods section to explain that the imaginary part of Δn was estimated from Ref. [46], found to have a negligible impact on the theoretical results and thus neglected in the modeling.

The authors describe the index perturbation of -0.25 as experimentally determined in the experimental results section. It would be interesting if the authors included how this experimentally determined number was measured, in the supplementary information section.

The real part of the refractive index change was determined from the shifting of the fringes in the transmission spectrum of an asymmetric Mach-Zehnder interferometer (MZI) when one of the waveguide arms was perturbed by the pump pulse. This investigation was reported in Ref. [6], where a relation between pump fluence and refractive index change is reported.

It may be worth noting that the evaluation of $\text{Im}(\Delta n)$ in a MZI based on Beer-Lambert law attenuation, owing to the fact that the perturbations are created in a narrow waveguide, includes not only absorption but also scattering by the perturbation. Thus, this value cannot be used to model perturbations in a uniform medium as in a MMI (hence the theoretical estimation discussed above).

Action taken: We now explain in the Methods section how the real part of the refractive index change was measured. As the same study was performed in a previous paper, we do not find it necessary to add a dedicated figure in the Supplementary Information.

The authors show that the measured perturbation maps are correlated most strongly for a different waveguide geometry as intended in the fabrication, suggesting that the delta is from fabrication variation which is confirmed with SEM. It would be very interesting if the authors plotted, in addition to the measured vs. modeled perturbation map, a measured vs. modeled vs modeled using the MMI parameters that have the highest correlation. This would especially be interesting in cases where there is most disagreement, such as the 3x3 MMI case outlined in Fig.4.

This is an interesting suggestion that we followed for the 3x3 MMI of Fig. 4. The theoretical perturbation maps using the MMI parameters from the SEM image or the MMI parameters yielding the highest correlations are indeed in better visual agreement with the measured perturbation map than the map computed using the MMI design parameters, as reported in Fig. 4.

Action taken: We added 2 new figures in the Supplementary Information (Figs. S6 and S7) that compare the measured perturbation map with the predicted perturbations maps using design, SEM and maximum correlation device parameters for the best and worst correlating port coupling, respectively, in the 3x3 MMI. In the main text, we added a sentence after the discussion on Fig. 4.

One of the motivations for this type of experimental analysis is characterization of optical components at the wafer-level in a manufacturing environment, as motivated by the authors in the introduction. Such a characterization would take time, which is a vital factor in enabling wafer-level deployment. It would be interesting if the authors described how much time a single device characterization consumed using their setup, and a rough estimate on how fast a commercial system could test a single device, to

help the reader understand the efficacy of this technique in a high-volume manufacturing.

We agree that this is a very useful information that should be discussed in the manuscript. Scan times depend on device size, scan step size, piezo scan speed and the integration time of the lock-in amplifier. With our setup, we achieved scan speeds on the order of a few $\mu\text{m}\cdot\text{s}^{-1}$, resulting in full scan times on the order of minutes. These scans can be further optimized for speed by adjusting certain parameters, such as reducing the integration time, at the expense, however, of signal-to-noise ratio.

In a commercial environment, a faster scanning system such as a mirror galvanometer would enable faster scan speeds in the order of $10\text{ m}\cdot\text{s}^{-1}$ while still achieving μm accuracy. Combined with faster optical modulators to improve dwell times to the region of $100\ \mu\text{s}$, scan times could be reduced to the order of a few seconds per device.

Action taken: We added several sentences in the before-the-last paragraph of the Discussion section to emphasize the importance of increasing characterization speed for deployment of the technique in industry.

ADDITIONAL CHANGES IN THE MAIN TEXT

- We have corrected a sentence in the introduction to clarify the validity of the perturbative approach used in previous studies based on the adjoint-variable method. The previous sentences were giving the wrong impression that Ref. [34] was less correct than other implementations of the method, while they are all identical in the approximations made (and would fail to recover as accurate results as our method, as shown in Fig. S1).
- We have rescaled Figures 2, 5 and 6 to make them single-column for aesthetic reasons.
- We have supplied a DOI for the dataset accompanying the manuscript.

REVIEWERS' COMMENTS:

Reviewer #1 (Remarks to the Author):

The authors have satisfactorily addressed my comments and suggestions. I recommend the paper is accepted for publication in Nature Communications.

Reviewer #3 (Remarks to the Author):

The authors have thoroughly addressed all comments and concerns, and there are no additional comments from the reviewer. It is recommended that this paper be accepted for publication.